# Fabrication of Large-Area Micro-Hexagonal Cube Corner Retroreflectors on Three-Linear-Axis Ultraprecision Lathes

**DOI:** 10.3390/mi14040752

**Published:** 2023-03-29

**Authors:** Senbin Xia, Ziqiang Yin, Cheng Huang, Songtao Meng

**Affiliations:** 1State Key Laboratory of Precision Electronic Manufacturing Technology and Equipment, Guangdong University of Technology, Guangzhou 510006, China; 2Guangdong Provincial Key Laboratory for Micro-Nano Manufacturing Technology and Equipment, School of Electro-Mechanical Engineering, Guangdong University of Technology, Guangzhou 510006, China

**Keywords:** hexagonal cube corner array, retroreflectors, diamond shifting cutting, ultraprecision machining

## Abstract

Hexagonal cube corner retroreflectors (HCCRs) are the micro-optics arrays with the highest reflectivity. However, these are composed of prismatic micro-cavities with sharp edges, and conventional diamond cutting is considered unmachinable. Besides, 3-linear-axis ultraprecision lathes were considered unfeasible to fabricate HCCRs due to the lack of a rotation axis. Therefore, a new machining method is proposed as a viable option to manufacture HCCRs on the 3-linear-axis ultraprecision lathes in this paper. For the mass production of HCCRs, the dedicated diamond tool is designed and optimized. The toolpaths are proposed and optimized to further increase tool life and machining efficiency. The Diamond Shifting Cutting (DSC) method is analyzed in-depth both theoretically and experimentally. By using the optimized methods, the large-area HCCRs with a structure size of 300 µm covering an area of 10 × 12 mm^2^ are successfully machined on 3-linear-axis ultraprecision lathes. The experimental results show that the whole array is highly uniform, and the surface roughness Sa of three cube corner facets is all less than 10 nm. More importantly, the machining time is reduced to 19 h, which is far less than the previous processing methods (95 h). This work will significantly reduce the production threshold and costs, which is important to promote the industrial application of HCCRs.

## 1. Introduction

The fabrication of an optical microstructure array has been the focus of many recent research projects, as it can be used to enhance various additional functions on technical surfaces, thus driving progress in many areas of application [1,2,3,4,5,6]. Retroreflectors are a special optical microstructure that allow the incident light to be reflected to their source, regardless of its incident direction. Due to their unique optical properties, retroreflectors are widely used in vehicle applications [7], pseudo-phase conjugate wavefront corrector [8,9], free-space optical communication [10], atmospheric sounding [11,12]

Two common types of retroreflectors are the cat’s eye and the cube corner. The former has a large receiving angle and large divergence characteristics, while the latter has a higher reflection efficiency [13]. The cube corner retroreflector will be discussed only in this paper because of its high reflectivity. According to the aperture shape of the cube corner, it can be divided into triangular and hexagonal cube corners, as shown in Figure 1. The triangular retroreflectors are only capable of retroreflecting 66.7% of the incident light. In contrast, hexagonal cube corner retroreflectors (HCCRs) can theoretically reflect up to 100% of the incident light [14].

HCCRs, however, are difficult to machine, because they are composed of prismatic micro-cavities with sharp edges. The pin-bundling-electroforming (PBE) technique has been widely employed to manufacture HCCRs [15]. However, the PBE technique is expensive to operate with a time-consuming multi-step machining process. Besides, it is suitable for fabricating millimeter-scale structures rather than micron-scale structures. Schönemann et al. proposed and developed Diamond Micro Chiseling (DMC) in 2008, 2012, 2014, and 2018 [6,16,17,18]. At least five numerically controlled axes (three linear and two rotational) are required for the DMC process. In addition, it is necessary to machine three triangular cube corners to fabricate one complete hexagonal cube corner. This method is a multiple and complex process, because the diamond tool needs to be multi-rotated and multi-repositioned, causing inefficient machining and increased machining errors, which is still a challenge for large-area array.

Sama Hussein et al. proposed the ultra-precision single-point inverted cut (USPIC) to machine the right triangular prism retroreflectors (RTP) in 2016 [19]. Milliken et al. developed it to an enhanced bidirectional USPIC in 2018 [20]. That is capable of overcoming the shortcomings of the previously employed cutting strategies. Although the RTP is a potential alternative to cube corner geometry, USPIC cannot fabricate the HCCRs. Milan et al. reported the micro-milling to manufacture the prismatic retroreflectors in 2020 [21]. However, it is difficult to fabricate the small-size microstructure because the micro-milling tool has a certain diameter. Our team has recently presented a new manufacturing technology named Diamond Shifting Cutting (DSC) in 2022 [22]. In the past, it has been successfully demonstrated on a small area with the single structure sizes ranging from micrometers to millimeters. In particular, DSC can improve the machining efficiency by at least 3 times compared with DMC process. It has shown the potential to be more suitable for the large-area production of HCCRs. However, the DSC process requires four numerically controlled axes (three linear and one rotational). Secondly, it has only been demonstrated on a small area and the surface roughness of the facet 3 is not good enough for the optical surface.

In summary, only two techniques have been reported to directly manufacture HCCRs. However, it still has the problem with time-consuming or difficult to achieve optical surface quality. Besides, the above methods require at least four numerically controlled axes for the machine tool to fabricate the HCCRs. For the three linear axis machine tool equipped with the *X*-, *Y*-, and *Z*-axes, it was considered infeasible to fabricate HCCRs due to the lack of rotation axis. As a result, the industrial application of HCCRs is greatly limited by the above problems.

Therefore, for promoting the industrial application of HCCRs, this work proposed a new machining method as a viable option to manufacture HCCRs on the 3-linear-axis ultraprecision lathes. By optimizing the tool geometry parameters and toolpaths generation, a large-area array of HCCRs can be machined.

## 2. Diamond Shifting Cutting Method on Three-Linear-Axis Ultraprecision Lathes

Hexagonal cube corner is a micro-concave structure with sharp edges, and it is considered impossible to manufacture by conventional machining methods. Currently, PBE process is difficult to fabricate HCCRs with small-sized structures, large-area array, and high precision, while DMC process requires multi-rotations and multi-repositioning of the diamond tool, thus resulting in low machining efficiency. More importantly, the fabrication of HCCRs on 3-axis machines has not been reported.

### 2.1. Analysis of the Structural Characteristics of HCCRs

Figure 2 illustrates the structural characteristics of HCCRs, and some interesting regularities are found. HCCRs consist of sequential concave elements, and each column or row is a spaced 1/2 element, which provides the possibility for continuous machining of the entire array. It should be noted that Facet 1 and Facet 2 of the hexagonal cube corner are located on Facet A, while Facet 3 is on Facet B, as shown in Figure 2b. Besides, an empty cavity with the angle of 90° is located between Faces A and B, and this phenomenon is arranged in a certain regular pattern in the whole array. Therefore, the V-groove-first strategy is established to remove a large amount of material, which provides a basis for the fabrication of HCCRs and also, cuts down a lot of machining time. Then, the hexagonal cube corner array can be fabricated on Facet A and B. It has significant implications for the fabrication of the large-area array. Therefore, diamond shifting cutting was developed.

As shown in Figure 2, it is assumed that *a* is the length of the hexagonal cube corner. According to the triangle principle, the angle *θ* is equal to *θ*_1_. In this case, the calculation of *θ* can be given by Equations (1)–(3):(1)$$l_{DE}=l_{AB}=a$$
(2)$$l_{DC}=l_{OC}/2=\sqrt{2}/2\times a$$
(3)$$\theta =\tan^{-1}\left(l_{DE}/l_{DC}\right)=\tan^{-1}\left(\sqrt{2}\right)\approx 54.74{}^{\circ}$$

### 2.2. Principle of Diamond Shifting Cutting Method

As shown in Figure 2b above, Facet A makes a fixed angle *θ* to the horizontal direction. In order to facilitate the design and mounting of subsequent diamond tools, the workpiece needs to be placed at an angle of *θ* in this paper. The geometry of the dedicated diamond tool needs to be designed based on the micro-cavity and V-groove; see Figure 3. The angle *β*_1_ of the diamond tool needs to be equal to the angle between Facet A and B, which generally is 90° [23], for the fabrication of the V-groove array. Two cutting edges of the diamond tool are required to generate the V-groove. In addition, Facet 1 and 2 of the cube corners are manufactured by the major cutting edge, while Facet 3 will be generated by the sub-cutting edge. Then, a micro-cavity will be fabricated as the diamond tool moves along the selected red path.

To illustrate the principle of DSC, the manufacturing process of HCCRs with a 3 × 3 array are used in this paper, as shown in Figure 4. DSC process will be divided into two parts to enable the fabrication of large-area HCCRs. A fixed angle *θ* should be maintained in the whole process.

The V-groove array will be manufactured first (see Figure 4a). The cutting process starts from point A, and the diamond tool moves along the [–X, +Y] direction to the next point (point B). Then, the diamond tool needs to further cut in the [+Y] direction to the point C. Finally, the diamond tool will be retracted in the [+X, +Y] direction. As a result, a complete V-groove is generated. The workpiece needs to be moved along the [+Z] direction in the distance of *a* to the next row. Repeat the above process until all of the V-groove array is manufactured. It is worth noting that this part is designed to remove a large amount of material, so the higher cutting speed and cutting depth are allowed to improve the machining efficiency.

After that is the fabrication of the hexagonal cube corner array. The material removal process is performed on Facet A of the V-groove array, as shown in Figure 4b. This machining process starts from the last row of the V-groove array to reduce the auxiliary cutting motion. The cutting motion starts from point 1 on Facet A, then goes along the [–X, +Y] direction to the valley of the cube corner (the point 2), and finally, in the [+X, +Y] direction to the end point (point 3). Thus, three facets of the hexagonal cube corner are generated in one process step. In order to machine the next row of the hexagonal cube corner, the workpiece needs to be moved along the [–Z] direction in the distance of *a*. More importantly, the diamond tool also needs to shift along the [+Y] direction in the distance of *S* to the new cutting point, as shown in the blue path, while in the yellow path, the direction needs to be changed to the [–Y] direction with the same distance. Thus, the HCCR with a 3 × 3 array is fabricated.

Assume that *ØD* is the circumcircle diameter of the hexagonal cube corner. The length of the hexagonal cube corner can be given by Equation (4). The shifting distance *S* can be calculated by Equation (5). According to the geometric characteristics of the hexagonal cube corner, the width (*w*) and depth (*d*) of the V-groove are equal to the length of the hexagonal cube corner (*a*) and the shifting distance (*S*), respectively.
(4)a=D/2×sin⁡(θ)
(5)$$S=a\times sin(\beta_{1}/2)$$

### 2.3. Machining System Configuration

Ultra-precision machines with three linear axes (without the rotary axes) are much lower in cost than the four- or five-axis machines with rotary axes. This is important for the industrial manufacturing of large-area HCCRs. Therefore, a DSC method based on three-axis linear motion is developed above. Crucially, a special fixture also must be designed for the DSC process on three-axis ultraprecision lathes. The dedicated fixture was machined with an inclined plane, and the inclined angle equal to the fixed angle *θ* (see Figure 3), and is generally designed as 54.74°, as shown in Figure 5.

The machine tool consists of three linear axes (*X*-, *Y*-, and *Z*-axis). The workpiece is mounted on the inclined plane of the dedicated fixture, and the dedicated fixture is set on the *Z*-axis, while the diamond tool is located on the *X*-axis and moves linearly with the *X*-axis and *Y*-axis.

## 3. Design and Optimization of Diamond Tool Geometry

### 3.1. Dedicated Diamond Tool for DSC

The DSC of HCCRs relies on the dedicated diamond tool, as shown in Figure 6. The diamond tool requires two cutting edges in the DSC process. The major cutting edge is used to create Facet A in the V-groove array and fabricate Facet 1 and 2 in the hexagonal cube corner array, while the sub-cutting edge is used to generate the rest facets. It can be seen that the major cutting edge has more cutting processes than the sub-cutting edge. Therefore, the tool edges are designed with the radius of *r_α_* = 1.2 µm and *r_β_* = 300 nm, respectively, to increase the tool life. To the fabrication of multi-size microstructures, the length of the cutting edge should be designed to be larger than the structures size of the hexagonal cube corner. The angle *β*_1_ between the two cutting edges is designed to be equal to the angle *δ*_2_ between the two reflecting facets (Facet 1 and Facet 3) of the hexagonal cube corner (see Figure 3) and generally is 90° in this paper.

### 3.2. Optimization of the Diamond Tool Geometry

The effect of the tool geometry on the tool life cannot be ignored, especially in the fabrication of a large-area array. Therefore, it is necessary to optimize the geometric parameters of diamond tools. In large-area machining, the wedge angle *β* is the main factor that affects the tool life, while the clearance angles *α*_1_ and *α*_2_ play an important role in avoiding interference with the machined surface. Therefore, four diamond tools were designed, and DSC experiments were conducted separately under the same conditions. The diamond tools are manufactured by the manual crystal diamond (MCD) and supplied by the SINJIN Diamond in Korea. The diamond tools are measured by the high precision optical microscope to evaluate the tool wear, as shown in Figure 7.

From Figure 7d, the sub-cutting edge of the No.1 diamond tool had no wear, and there was also no built-up edge (BUE) on the rake face and the clearance face. However, the major cutting edge had obvious wear. In this case, it will result in higher surface roughness, as shown in Figure 7b. This indicates that the wedge angle *β* is not enough for the fabrication of large-area HCCRs. As the *β* increases to 55°, minor wear occurs at the sharp corners of the major cutting edge and the sub-cutting edge of the No.2 diamond tool, illustrated in Figure 7e. For the No.3, although the cutting edges were not worn, a large amount of work material stuck to the cutting tool, as shown in Figure 7f. This built-up layer was considered as a protective layer, which can prevent the rake and clearance face from wear. However, Facet 3 will be destroyed by this excessive material sticking to the clearance face, as shown in Figure 7c. It indicates that the wedge angle *β* is enough for the DSC process. However, the clearance angle *α*_2_ needs to be increased to protect Facet 3. Therefore, after the further experiment, the final diamond tool (No.4) was identified, and it showed an excellent performance, with neither significant wear on the two cutting edges nor BUE on the rake and clearance face; see Figure 7g.

## 4. Toolpath Generation and Optimization

Tool path generation is a key part of the machining process. Industrial manufacturing of the large-area array is sensitive to the machining time, and it is significantly affected by the toolpath generation. In the manufacturing of HCCRs by DSC process, the toolpath can be divided into two groups. The first group is the fabrication of the V-grooves array. The cutting strategy with larger cutting speed and cutting depths is selected to remove a large amount of material in this process. Another group is the fabrication of the hexagonal cube corner array. Lower cutting speed is required, because the speed direction will be changed periodically during this machining process. As a result, most machining time will be taken. Therefore, the toolpath generation needs to be optimized in this process. Three feasible toolpaths generation are proposed in this paper, as shown in Figure 8.

The first approach is the single microstructure cutting, as shown in Figure 8a. A hexagonal cube corner will be individually and completely machined, then turned to the next one. In this case, more auxiliary motions are required to retract and return the diamond tool, so that the machining time will be increased due to more auxiliary motions. The second approach is modified by the first approach, as shown in Figure 8b. The microstructure of the same row will all be generated in the last cutting, which will greatly improve the consistency of the entire array, while the machining time is not significantly reduced compared with the first method.

Both methods above are viable to manufacture HCCRs, but there is still a lot of unnecessary auxiliary motion, which will increase the machining time, especially in the fabrication of a large-area array. Besides, it is worth noting that both toolpaths above are always in a bigger negative rake angle at cutting Facet 2, as shown in the red toolpath in Figure 8, which will have a negative impact on Facet 2 and also, reduce the tool life.

Therefore, a third approach is proposed and selected in this paper. As can be noticed, this toolpath has the characteristics of continuous cutting and layer-by-layer reduction of the maximum cutting depth (*ap*_1_ > *ap*_2_ > … > *ap_n_*). In this case, unnecessary auxiliary motions are avoided to greatly reduce the machining time, and the consistency of the entire array is improved. In addition, layer-by-layer reduction of the maximum cutting depth can avoid the linear increase of the removal area with the increase of the cutting layer to protect the diamond tool. Lastly, the angle of the negative rake angle process is variable, and only reaches its maximum when approaching the theoretical size of the structure. This is much lower than the other approach mentioned above and it is significant for improving the tool life and Facet 2.

The method of gradually decreasing the maximum depth of cut can be assumed to be an equivariant sequence. The first *ap*_1_ and last layer apn need to be given, and the other *ap_i_*s can be obtained by the following Equations (6)–(8), where *n* is the total cutting layer of a cube corner and ∆ is the tolerance of the arithmetic sequence:(6)$$n=\left(2\times d\right)/\left({ap}_{1}+{ap}_{n}\right)$$
(7)$$\Delta =\left({ap}_{1}-{ap}_{n}\right)/\left(n-1\right)$$
(8)$${ap}_{i}={ap}_{1}-(i-1)\times \Delta$$

## 5. Experiment Setup

### 5.1. Machine Tool Setup

The machining experiments are conducted on three-axis ultraprecision lathes, and they consist of three linear axes (*X*, *Y*, *Z*), as shown in Figure 9. The *X*-axis, *Y*-axis, and *Z*-axis are 300 mm, 300 mm, and 150 mm, respectively. Cutting fluids are used to improve the tool life and bring the cutting heat out. The optical system is applied to monitor the machining process. The spectral confocal sensor (resolution of 3 nm, maximum sampling frequency of 66 kHz) is used to measure the installation error of the dedicated fixture, to make sure the machine setup error is below 0.01°.

### 5.2. Experiment Preparations

The No.4 diamond tool is selected to conduct the experimental studies, which was optimized in Section 3.2. The tool geometries are shown in Figure 7g. The third toolpath generation is chosen to the following experiment. It should be pointed out that DSC is a new machining method that we developed. In the fabrication of the hexagonal cube corner array, the cutting direction will change periodically. Higher cutting speed will inevitably cause the vibration of the machine tool. In addition to the cutting speed, the workpiece material is also an essential factor that directly affects the surface quality. For this reason, this paper conducted exploratory research on the cutting speed and different workpiece materials to shorten the machining time and verify their influence on the DSC process through a series of experiments. Aluminum alloy 6061 (Al6061) and Brass are selected as the experimental materials due to their good cutting performance, and the machining parameters are listed in Table 1 and Table 2.

## 6. Results and Discussion

### 6.1. Effects of Workpiece Material and Cutting Speed

For evaluating the performance of the DSC process, the hexagonal cube corner with 3 × 6 array was machined in different workpiece materials while varying the cutting speed of the process. For determining the process performance, the surface roughness (Sa) was measured by an optical profiler (Bruker GT-X). The scanning electron microscope (SEM) was used to evaluate the integrity of the processing of HCCRs. The experimental scheme and the measure results are drawn in Figure 10.

From Figure 10, the Al6061 sample shows a better cutting performance with lower machined surface roughness than the Brass sample at the same cutting conditions. In particular, the best achievable surface roughness of all three reflective facets on the Al6061 sample was measured as to less than 10 nm at the same time, while the brass sample also yielded no optical surface finish at the same cutting conditions. It means that Al6061 is more suitable to manufacturing HCCRs by the DSC method.

The cutting speed shows a significant effect on all the three reflective surfaces of the cube corner in the DSC process. The surface roughness (Sa) varies within a small range, when the cutting speed is lower than 10 mm/min. However, it increases sharply when the cutting speed exceeds 25 mm/min. It indicates that when the cutting speed is higher than that abrupt change point, the machining surface quality will deteriorate sharply. Besides, the effect of cutting speed on the surface roughness shows differently on the three facets. Cutting speed has the greatest effect on Facet 1 and 2, while it has the least effect on Facet 3.

For further analysis, the machined facets of Al6061 were measured by SEM and the optical profiler, and the measured results are shown in Figure 11. Three reflective facets show significant defects at high speeds (40 mm/min and 80 mm/min), as shown in Figure 11a,b. The reason for these defects is the change of cutting direction, as shown in Figure 10a. Especially at such high cutting speeds, the abrupt change in cutting directions results in a significant jerking, causing the machine tool to vibrate. This vibration will be transferred to the machined surface through the diamond tool, resulting in a sharp increase in surface roughness, as shown in the morphology of Figure 11, while under low-speed cutting (less than 10 mm/min), the three cutting surfaces show excellent performance, as shown in Figure 11c. In addition, the reason for the different effects of the abrupt change in cutting directions on the three surfaces is that the vibration caused at a high speed occurs mainly in the direction parallel to the cutting speed, which results in a higher surface roughness on Facet 1 and 2 than on Facet 3.

Therefore, although a higher cutting speed is significant to improve the machining efficiency, the cutting speed should be selected reasonably to avoid the vibration caused by the abrupt change in cutting directions. In the DSC mode, the cutting speed should be set below 10 mm/min to manufacture HCCRs.

### 6.2. Machining of Large-Area Hexagonal Cube Corner Array

Using optimization strategies described above, it is possible to machine a highly efficient large-area array with a structure size of 300 µm covering an area of 10 × 12 mm^2^ on the Al6061 sample. The optimized diamond tool (No.4) and the third toolpath generation are selected. From Figure 10, the cube corner has the best surface roughness at a cutting speed of 5 mm/min; the cutting speed is selected in the experiment. The other machining parameters are shown in Table 1 and Table 2.

The SEM image is illustrated in Figure 12. The entire functional surface shows as highly uniform, and the protruding edges of the cube corners are sharp and straight over the whole array. For further evaluation, the six vertices of the cube corner element were measured using a high-precision optical microscope; the measured results of the individual cube corners are in agreement with the theoretical values (*ØD*), and the machining error is controlled within the range of −1.1~2.3 µm. The surface roughness is measured with the help of a special fixture (see Figure 5), and the reflective surface can be measured for every 120° of workpiece rotation. The experiment results show that the average surface roughness (Sa) of the three reflecting surfaces are 7.7 nm, 9.5 nm, and 8.3 nm, respectively. All the facets of the hexagonal cube corner are suitable for optical application. It indicates that it is feasible to manufacture large-area HCCRs on the 3-axis machine by the DSC method.

Machining time is also an important criterion to evaluate the machining method, especially in machining large-area microstructure array. After all the preliminary preparations were completed, the machining time of manufacturing HCCRs with an area of 10 × 12 mm^2^ was recorded. The machining time was measured to approximately 19 h in this paper, which is far less than the DMC method.

## 7. Conclusions

In this paper, we presented a novel method for machining large-area HCCRs with a three-linear-axis ultraprecision lathe, which is still the only process available to fulfill that task. The diamond tool is dedicatedly designed and optimized, and the toolpaths generation is proposed and further optimized. The main factors affecting the machined surface are analyzed. The experiment is conducted to machine a large-area hexagonal cube corner array. The important conclusions can be summarized as follows:

(1) HCCRs can be directly fabricated on the 3-linear-axis ultraprecision lathe by the dedicated fixture with an inclined plane. To enable mass manufacturing, the V-groove-first strategy is established to greatly reduce the machining time, and the diamond tool is designed and optimized. The special toolpaths are proposed and optimized to further increase the tool life and the machining efficiency.

(2) The work material and cutting speed affecting the optical surface quality are analyzed. The experimental results show that the Al6061 sample shows a better cutting performance with lower machined surface roughness than the Brass. In addition, an abrupt change point in cutting speed (25 mm/min) is found. Above the determined abrupt change point, the machine tool will generate the significant jerking, resulting in a sharp increase in surface roughness, while under low-speed cutting (less than 10 mm/min), DSC can achieve the optical surface quality cutting.

(3) The large-area HCCRs with the size of 300 µm covering an area of 10 × 12 mm^2^ is successfully machined on the Al6061 sample. The measure results show that the machining error of the structure size is controlled to below 2.3 µm. The average surface roughness (Sa) of the three reflecting facets are 7.7 nm, 9.5 nm, and 8.3 nm, respectively. In particular, the machining time is recorded to approximately 19 h in this paper, which is far less than the previous processing methods (95 h).

## Figures and Tables

**Figure 1 micromachines-14-00752-f001:**
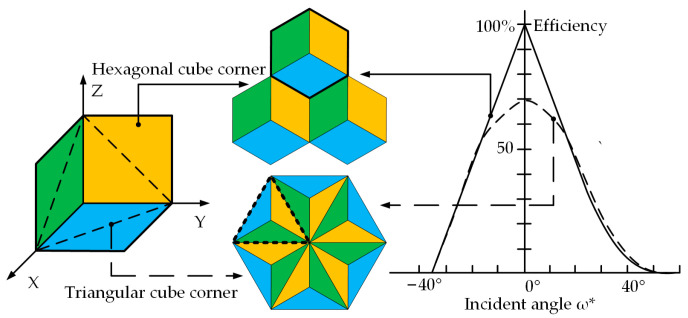
Two common types of retroreflectors.

**Figure 2 micromachines-14-00752-f002:**
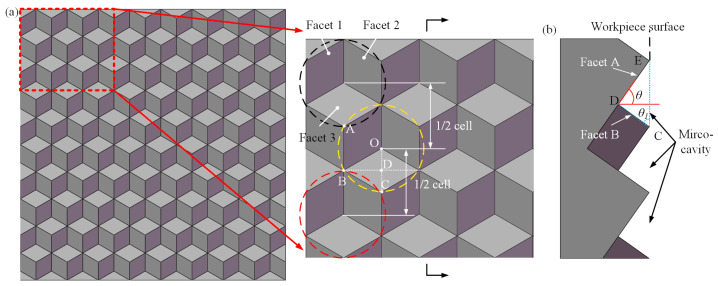
The structural characteristics of HCCRs. (**a**) Arrangement characteristics of the hexagonal cube corner arrays. (**b**) Profile view of HCCRs.

**Figure 3 micromachines-14-00752-f003:**
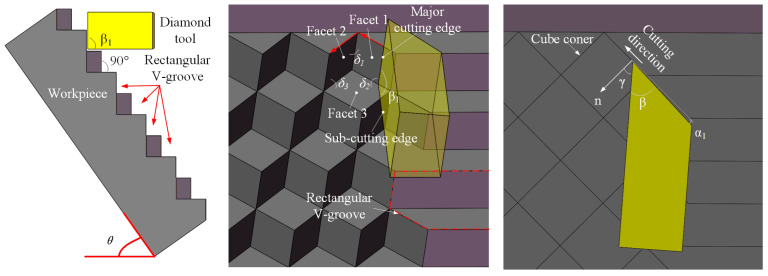
Relationship between diamond tool geometry and cube corner.

**Figure 4 micromachines-14-00752-f004:**
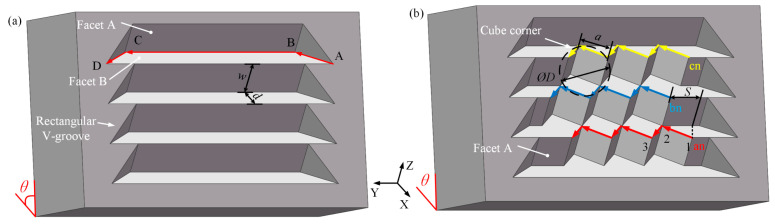
The principle of the DSC method. (**a**) Manufacturing method of V–groove array. (**b**) The fabrication of the hexagonal cube corner array.

**Figure 5 micromachines-14-00752-f005:**
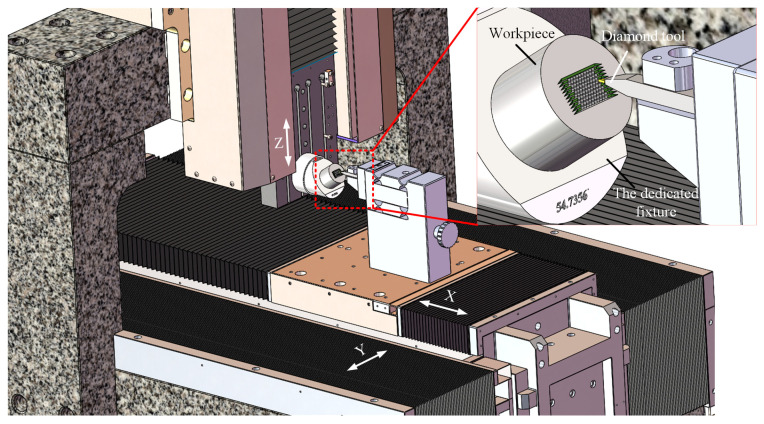
Configuration of three-axis ultraprecision lathes.

**Figure 6 micromachines-14-00752-f006:**
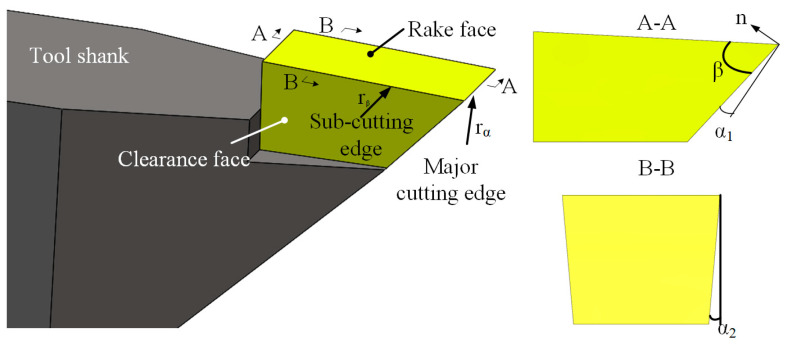
DSC tool and schematics of tool geometry.

**Figure 7 micromachines-14-00752-f007:**
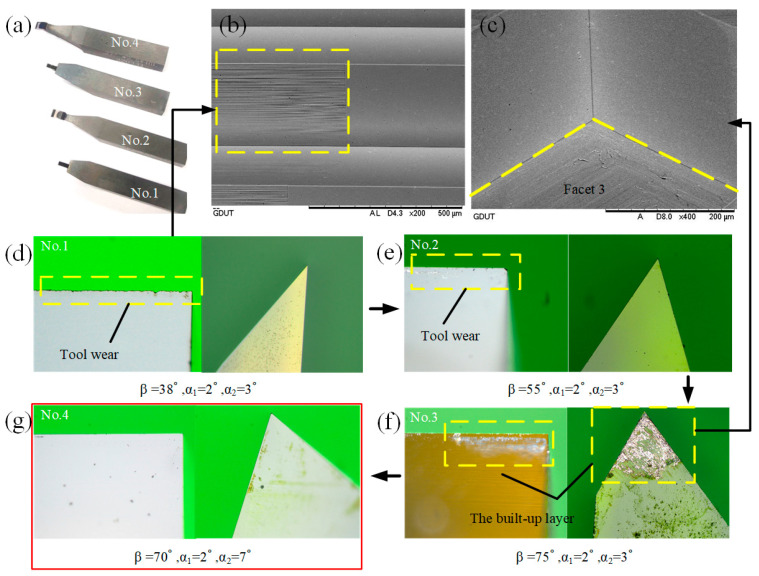
Tool wear for different diamond tools. (**a**) Four designed diamond tools. (**b**) Effect of tool wear on machined surfaces. (**c**) Effect of the built-up layer on machined surfaces. (**d**) The No.1 diamond tool after machining. (**e**) The No.2 diamond tool after machining. (**f**) The No.3 diamond tool after machining. (**g**) The No.4 diamond tool after machining.

**Figure 8 micromachines-14-00752-f008:**
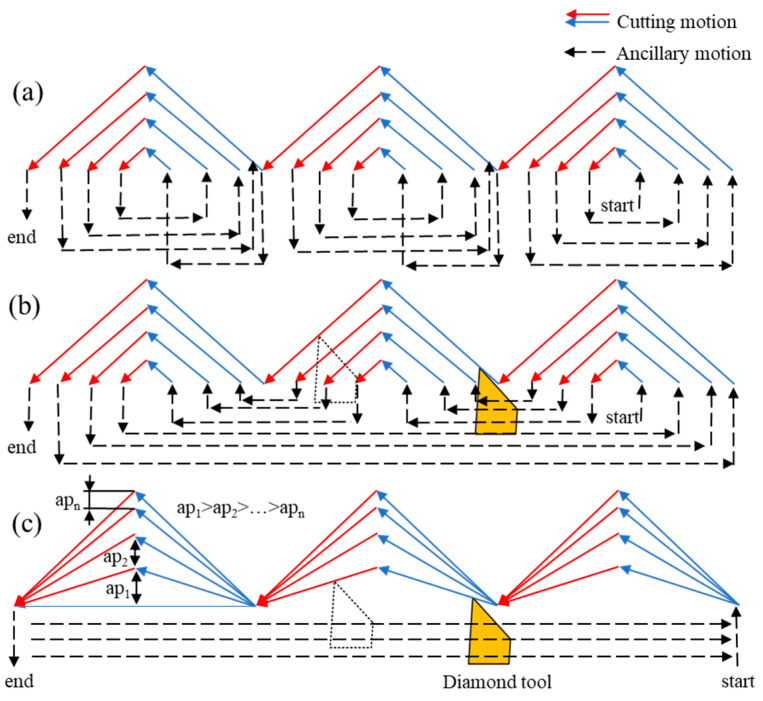
Optimization of toolpath generation. (**a**) Toolpath for the single microstructure cutting. (**b**) The modified approach by the first toolpath. (**c**) The optimized toolpath.

**Figure 9 micromachines-14-00752-f009:**
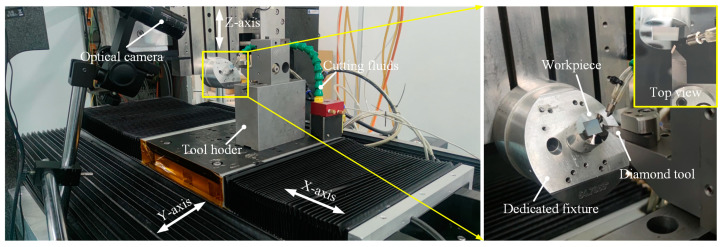
Three-axis ultraprecision lathes setup for the fabrication of HCCRs.

**Figure 10 micromachines-14-00752-f010:**
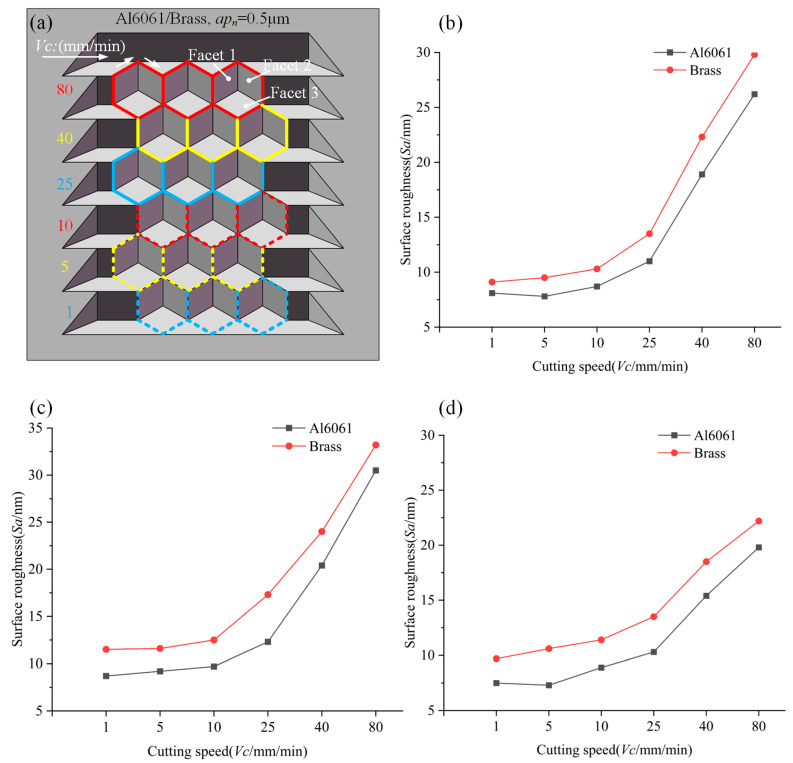
Effects of workpiece material and cutting speed. (**a**) Experimental scheme. (**b**) Effect of cutting speed on Facet 1. (**c**) Effect of cutting speed on Facet 2. (**d**) Effect of cutting speed on Facet 3.

**Figure 11 micromachines-14-00752-f011:**
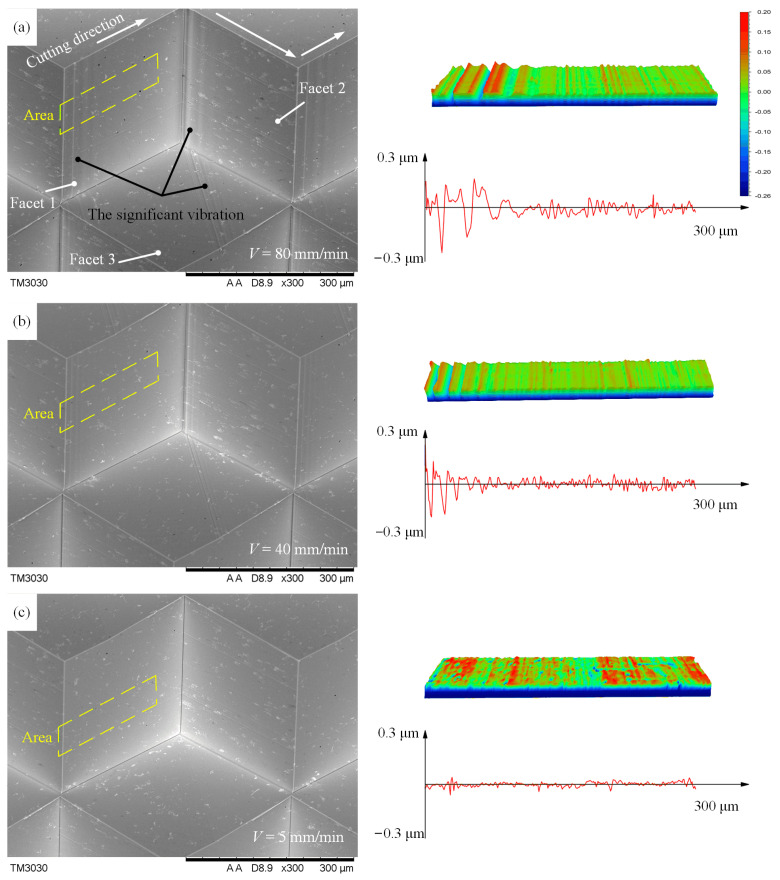
Effects of the abrupt change in cutting directions at different cutting speed on surface generation. (**a**) 80 mm/min. (**b**) 40 mm/min. (**c**) 5 mm/min.

**Figure 12 micromachines-14-00752-f012:**
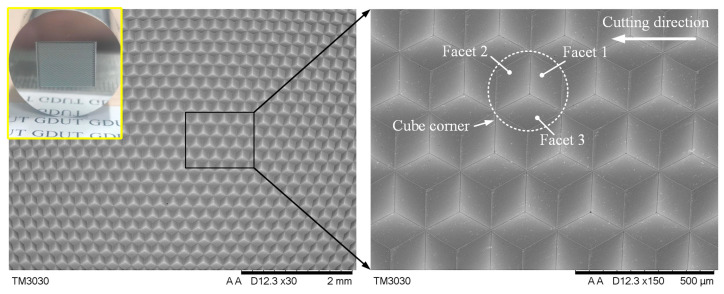
Ten mm × 12 mm cube corner array with a structure size of 300 µm.

**Table 1 micromachines-14-00752-t001:** The machining parameters of rectangular V-groove arrays.

Parameters	Rough Machining Process	Finish Machining Process
Depth of cut	5 µm	0.5 µm
Cutting speed	350 mm/min	10 mm/min

**Table 2 micromachines-14-00752-t002:** The machining parameters of hexagonal cube corner arrays.

Parameters	Rough Machining Process	Finish Machining Process
Max cutting depth *ap*_1_	5 µm	/
Cutting speed	40 mm/min	/
Min cutting depth *ap_n_*	/	0.5 µm
Cutting speed	/	1, 5, 10, 25, 40, 80 mm/min

## Data Availability

All data needed to evaluate the conclusions in the paper are provided in the paper. Additional data related to this paper may be requested from the authors.

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
