# Peer review of "Fabrication of Large-Area Micro-Hexagonal Cube Corner Retroreflectors on Three-Linear-Axis Ultraprecision Lathes"

_micromachines, 2023, doi:10.3390/mi14040752_

Round 1

Reviewer 1 Report

The research content of this paper is rich and interesting. In the present work, authors investigated a new machining method as a viable option to manufacture HCCRs on the 3-linear-axis ultraprecision lathes There are some questions and considerable points that should be elucidated: 

1.      Line 18- DSC full form is required.

2.       Line 107,109,157 -Theta value (54.74°) calculations are required. Represent theta calculations, as it is dependent on the angle between Facet A between Facet B. 

3.      Line 111, 112- Show literature for design value of angle (90°) b/w facet A and facet B as Tool angle Beta1 is dependent (modification in the diamond tool is not an easy part and costly)

4.       In Figure 3 and Figure 6 the Same angle is denoted by Beta and Beta2, correction is required. 

5.      Characterization of Diamond tool waviness over the major-cutting edge and minor-cutting edge is required which can directly affect the roughness and profile of Facets. Elucidate angle β2 measurements in terms of Deviation and tolerance. 

6.      Line 171- Clarification required for selecting specific values for radius α and radius β.

7.      In Diamond tool geometry optimization, four tools were selected with different wedge angle (β). As the angle b/w facet A and facet B is 90°, so to maintain uniformity of rake angle in all trials β + α must be below 45° or either change in rake angle also can affect the comparison results. How same rake angle is maintained during comparison studies in all four different tools (As some high values for β i.e.-70°,75° is also considered in study)

Reviewer 2 Report

1. There are still some grammatical errors that shoul be checked, for example, BUE is generally short for built up edge, rather than BUP;

2. Some detail information on the accuracy of the machine setup, the provider of the diamond tool and the properties of workpiece should be provided;

3. 3D surface topography of the fabricated large-area hexagonal cube corner arrays should be provided to better illustrate the results.
